# LEARNING AND STEERING GAME DYNAMICS TOWARDS DESIRABLE OUTCOMES

## ABSTRACT

Game dynamics, which describe how agents' strategies evolve over time based on past interactions, can exhibit a variety of undesirable behaviours including convergence to suboptimal equilibria, cycling, and chaos. While central planners can employ incentives to mitigate such behaviors and steer game dynamics towards desirable outcomes, the effectiveness of such interventions critically relies on accurately predicting agents' responses to these incentives—a task made particularly challenging when the underlying dynamics are unknown and observations are limited. To address this challenge, this work introduces the Side Information Assisted Regression with Model Predictive Control (SIAR-MPC) framework. We extend the recently introduced SIAR method to incorporate the effect of control, enabling it to utilize side-information constraints inherent to game-theoretic applications to model agents' responses to incentives from scarce data. MPC then leverages this model to implement adaptive incentive adjustments. Our experiments demonstrate the efficiency of SIAR-MPC in guiding systems towards socially optimal equilibria, stabilizing chaotic and cycling behaviors. Comparative analyses in data-scarce settings show SIAR-MPC's superior performance compared to pairing MPC with state-of-the-art alternatives like Sparse Identification of Nonlinear Dynamics (SINDy) and Physics Informed Neural Networks (PINNs).

## 1 INTRODUCTION

Game theory provides a mathematical framework for studying strategic interactions among self-interested decision-making agents, i.e., players. The Nash equilibrium (NE) is the central solution concept in game theory, describing a state where no player has an incentive to deviate (Nash, 1950). Over time, research has shifted from simply assuming that an NE exists and players will eventually play it, to understanding *how* equilibrium is reached (Smale, 1976; Papadimitriou & Piliouras, 2019). This shift has led to a focus on *learning* in games, exploring how strategies evolve over time based on past outcomes, adopting a dynamical systems perspective (Fudenberg & Levine, 1998; Sandholm, 2010). It has been shown that game dynamics do not necessarily converge to NE but instead can display a variety of undesirable behaviors, including cycling, chaos, Poincaré recurrence, or convergence to suboptimal equilibria (Hart & Mas-Colell, 2003; Sato et al., 2002; Mertikopoulos et al., 2018; Milionis et al., 2023). Motivated by these challenges, our primary objective in this work is to determine:

*Can we steer game dynamics towards desirable outcomes?*

To address this problem, we adopt the perspective of a central planner who seeks to influence player behaviour by designing incentives. Our goal is to achieve this with minimal effort, ensuring that the incentives are both cost-effective and efficient. More importantly, we operate in a setting with unknown game dynamics and limited observational data, reflecting real-world scenarios where information is often incomplete or uncertain. To tackle these challenges, we introduce a new computational framework called Side Information Assisted Regression with Model Predictive Control (SIAR-MPC), designed to steer game dynamics by integrating cutting-edge techniques for real-time system identification and control. In the system identification step, we predict agents' reactions to incentives, which is especially challenging for settings where observational data is limited, difficult to obtain, or costly. To address this problem, we extend the recently introduced SIAR method

(Sakos et al., 2023), which was originally developed to identify agents' learning dynamics from a short burst of a system trajectory. To compensate for the absence of data, SIAR searches for polynomial regressors that approximate the dynamics, satisfying side-information constraints native to game theoretical applications. To adapt it to our needs, we broaden the scope of SIAR to incorporate the influence of control parameters and enable it to model the controlled dynamics, resulting in SIAR with control (SIARc).

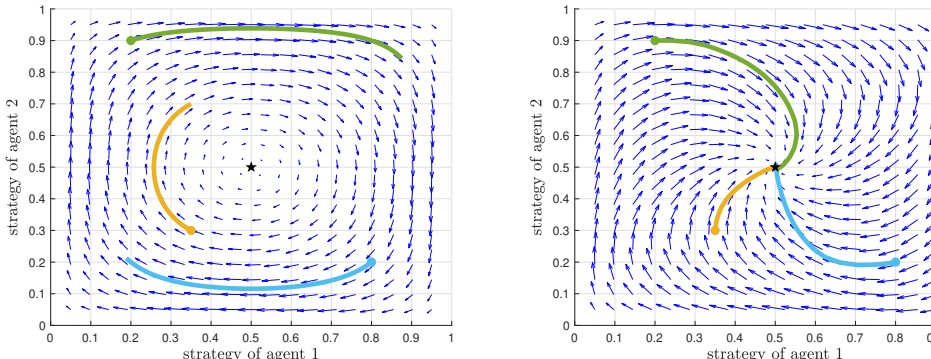

Figure 1: Replicator dynamics trajectories in the matching pennies game with and without control. Starting from three initial conditions, *(left)* without control the system cycles around the equilibrium; *(right)* with control, trajectories are guided towards the specified equilibrium (indicated by ⋆.)

Once agent responses to incentives are modeled with SIARc, we use MPC in the subsequent control step to develop a dynamic incentive scheme that steers the system towards desirable outcomes (see Fig. 1 for an example illustrating the impact of control). MPC is a control technique that leverages a mathematical model of the system to predict future behavior and calculate optimal control inputs that minimize a given objective function (Camacho & Bordons, 2007). A key advantage of MPC is its ability to handle input constraints, which is particularly relevant in our context since there are practical limits to the incentives that can be offered to agents. However, the effectiveness of MPC depends on having an accurate system model, highlighting the importance of the system identification step for its successful application.

Our key contributions are:

- **Framework for steering game dynamics:** We introduce the SIAR-MPC framework for steering game dynamics towards desirable outcomes when the underlying agent behaviours are unknown and observational data are scarce.

- **Demonstration of performance across diverse game types:** We demonstrate the effectiveness of our framework across a diverse range of game types, from zero-sum games like Matching Pennies and Rock-Paper-Scissors to coordination games such as Stag Hunt. Our experimental results show that SIAR-MPC can successfully steer system dynamics towards socially optimal equilibria and stabilize chaotic and cycling learning dynamics.

- **Superior performance in data-scarce settings:** We benchmark SIAR-MPC against an unconstrained regression method, Sparse Identification of Nonlinear Dynamics with control (SINDYc), as well as Physics-Informed Neural Network (PINN) coupled with MPC. Our results demonstrate that SIAR-MPC consistently achieves superior convergence and reduced control costs in data-scarce settings.

## 2    RELATED WORK

**Control of Game Dynamics:** Recently, incentive-based control has been applied extensively in multi-player environments (see, e.g., Riehl et al. (2018) and references therein). In addition, optimal control solutions have been given for specific evolutionary games and dynamics (Paarporn et al., 2018; Gong et al., 2022; Martins et al., 2023). However, in the aforementioned works the player behavior is known, and to the best of our knowledge a setup where the game dynamics are not a

priori known to the controller has only been explored in the recent works of Zhang et al. (2024) and Huang et al. (2024). Zhang et al. (2024) study the problem of steering no-regret agents in normal- and extensive-form games, under both full and bandit feedback. However, in their setup the agents are adversarial and not bound to fixed dynamics, making their setup incompatible with system identification. On the other hand, Huang et al. (2024) study this problem in the richer environment of Markov games under the additional assumption that the players' dynamics belong to some known *finite* class. This assumption allows the controller to utilize simulators of the class of dynamics to optimize for controls that identify the exact update rule with high probability. In contrast, SIAR-MPC does not rely on this assumption and instead takes advantage of the approximation guarantees of polynomial regression to acquire an accurate representation of an unknown system.

**System Identification:** In the last decades, the overall success of data-driven approaches in the scientific communities has inspired the development of Deep Learning (DL) architectures focused on the discovery of dynamical systems, e.g., Neural ODEs (Lu et al., 2018). More recently, increasing demand for more interpretable "black-box" models led to the development of various specialized architectures including Physics Informed Neural Networks (PINNs) (Raissi et al., 2019), Kolmogorov-Arnold Network ODEs (KAN-ODEs) (Koenig et al., 2024), Learning Across Dynamical Systems (LEADS) (Yin et al., 2021), and Deep Projection Networks (DPNets) (Kostic et al., 2024). Among the aforementioned data-driven technologies, we single out PINNs, and PINN-based architectures (Geneva & Zabaras, 2020; Ren et al., 2022; Lu et al., 2021), due to their ability to incorporate side information, e.g., physical laws, in their training process. This allows PINNs to also be used in data-scarce environments provided that the amount of side information available is adequate.

Two recent non-DL system identification frameworks that account for data scarcity are Sparse Identification of Nonlinear Dynamics (SINDy) (Brunton et al., 2016) and the work of Ahmadi & Khadir (2023), which rely respectively on Sparse-Polynomial Regression and Sum-of-Squares Optimization. SINDy is a sparsity-promoting method initially developed for the discovery of ODEs which has been extended through different approaches (Rudy et al., 2017; Mangan et al., 2016; Quade et al., 2018; Chu & Hayashibe, 2020) and applied extensively. An important extension of SINDy is SINDy with control (SINDYc) which incorporates the effects of control inputs (Kaiser et al., 2018). However, SINDy-based frameworks only allow for the incorporation of linear constraints with the notable exception of SINDy-SI (Machado & Jones, 2024), which allows for the integration of polynomial nonnegativity constraints based on the novel framework of Ahmadi & Khadir (2023). Even so, SINDy-SI is not tailored for the discovery of game dynamics. On the other hand, the recently introduced Side Information Assisted Regression (SIAR) (Sakos et al., 2023) method extends the work of Ahmadi & Khadir (2023) to games, incorporating various well-studied game-theoretic properties as polynomial nonnegativity constraints on the dynamics. At a technical level, both SINDy-SI and SIAR solve a hierarchy of semidefinite problems. In this work, we follow this approach and develop the SIAR-MPC framework with a focus on the identification and *control* of game dynamics through semidefinite programming in a data-scarce environment.

## 3 PRELIMINARIES

In this work, we model a multi-agent system as a time-evolving normal-form game of $n$ players. Each player $i$ is equipped with a finite set of strategies $\mathcal{A}_i$ of size $m_i$, and a time-varying reward function $u_i : \mathcal{A} \times \Omega \to \mathbb{R}$, where $\mathcal{A} \equiv \prod_{i=1}^n \mathcal{A}_i$ is the game's strategy space, of the form

$$u_i(a, \omega(t)) = u_i(a, 0) + \omega_{i,a}(t), \text{ for } t \in \mathbb{R}_+, a \in \mathcal{A}. \tag{1}$$

The value $\omega_{i,a}(t)$ denotes the control signal from the policy maker towards player $i$ regarding the strategy profile $a := (a_1, \ldots, a_n)$ at time $t$, where $a_i \in \mathcal{A}_i$. We refer to the ensemble $\omega_i(t) := \left(\omega_{i,a}(t)\right)_{a \in \mathcal{A}}$ as the control signal of $i$ at time $t$, and to $\omega(t) := \left(\omega_i(t), \ldots, \omega_n(t)\right)$ as the system's control signal at $t$. Typically, we restrict the control signals $\omega_i(t)$ in some semialgebraic sets $\Omega_i$. We are going to refer to the product $\Omega \equiv \prod_{i=1}^n \Omega_i$ as the game's control space. The value $u_i(a) := u_i(a, 0)$ describes the reward of player $i$ at strategy profile $a$ in the absence of any control by the policy maker. The utilities $u_i(\cdot)$, $i = 1, \ldots, n$ will be considered common knowledge throughout this work.

In addition to the above, each player $i$ is allowed access to a set of mixed strategies $\mathcal{X}_i \equiv \Delta(\mathcal{A}_i)$, which is the $(m_i - 1)$-simplex that corresponds to the set of distributions over the pure strategies $\mathcal{A}_i$

of $i$. The players' reward function naturally extends to the space of mixed strategy profiles $\mathcal{X} \equiv \prod_{i=1}^{n} \mathcal{X}_i$ with $u_i(x, \omega) = \mathbb{E}[u_i(a, \omega)]$ for all $x \in \mathcal{X}$ and $\omega \in \Omega$, where the expectation is taken with respect to the distributions $x_1, \ldots, x_n$ over the player's pure strategies.

Finally, we assume that the evolution of the above game is dictated by some controlled learning dynamics of the form

$$
\begin{aligned}
\dot{x}(t) &= f(x(t), \omega(t)) \quad \text{for } t \in \mathbb{R}_+, \\
x(0) &\in \mathcal{X},
\end{aligned}
\tag{2}
$$

where the update policies $f_i : \mathcal{X} \times \Omega_i \to \mathbb{R}^{m_i}$, $i = 1, \ldots, n$ and the ensemble thereof, given by $f(x, \omega) := (f_1(x, \omega_1), \ldots, f_n(x, \omega_n))$, are considered unknown, and are going to be discovered in the identification step of the framework described below. Notice that the above assumption also implies that, at each time $t$, the control signal $\omega_i(t)$ of player $i$ is observed by that player, while the strategy profile $x(t)$ is observed by all the players.

Throughout this work, given some strategy profile $x$, we adopt the common game-theoretic short-hand $(x_i, x_{-i})$ to distinguish between the strategy of player $i$ and the strategies of the other players. Furthermore, if $x_i$ corresponds to a pure strategy $a_i$ of $i$, we write $(a_i, x_{-i})$ to point to that fact.

## 4 THE SIAR-MPC FRAMEWORK

In this section, we describe the SIAR-MPC framework for the real-time identification and control of game dynamics. As outlined in the introduction, SIAR-MPC involves two steps. First, the system identification step aims to approximate the controlled dynamics in (2) using only a limited number of samples. Second, once the agents' reactions to payoffs are modeled, the control step employs MPC to steer the system towards a desirable outcome by optimizing specific objectives. The following subsections details these two steps.

### 4.1 THE SYSTEM IDENTIFICATION STEP

To model the controlled dynamics in (2) we extend the SIAR framework introduced in Sakos et al. (2023), which was in turn motivated by recent results in data-scarce system identification (Ahmadi & Khadir, 2023). SIAR relies on polynomial regression to approximate agents' learning dynamics of the form $\dot{x}(t) = f(x(t))$ based on a small number of potentially noisy observations $x(t_k), \dot{x}(t_k)$ (typically, $K = 5$ samples) taken along a short burst of a single system trajectory. To ensure the accuracy of the derived system model, SIAR searches for polynomial regressors that satisfy side-information constraints native to game-theoretic applications, which serve as a regularization mechanism.

For our control-oriented scenario, we extend the SIAR method to account for the influence of the control signal $\omega(t)$. We refer to this extended method as SIAR with control (SIARc). The aim of SIARc is to model the controlled dynamics in (2) for each agent $i$ through a polynomial vector field $p_i(x, \omega)$. To do so, during the data collection phase, we assemble a dataset $x(t_k), \omega(t_k), \dot{x}(t_k)$, where $x(t_k)$ represents a snapshot of the system state, $\omega(t_k)$ is a randomly (typically, normally distributed) generated input reflecting various possible incentives given to players (cf. equation 1), and $\dot{x}(t_k)$ is the velocity at time $t_k$, which is obtained either through direct measurement (if possible) or estimated from the state variables. The process of training the SIARc model essentially involves solving an optimization problem to find a polynomial vector field that minimizes the mean square error relative to this dataset. However, straightforward regression often yields suboptimal models due to the limited available samples. To overcome this challenge, we search over regressors that satisfy additional side-information constraints, encapsulating essential game-theoretic application features and refining the search for applicable models. Formally, a generic SIARc instance is given by

$$
\begin{aligned}
\min_{p_1, \ldots, p_n} \quad & \sum_{k=1}^{K} \sum_{i=1}^{n} \left\| p_i(x(t_k), \omega(t_k)) - \dot{x}_i(t_k) \right\|^2 \\
\text{s.t.} \quad & p_i \text{ are polynomial vector fields in } x \text{ and } \omega \\
& p_i \text{ satisfy side-information constraints.}
\end{aligned}
\tag{3}
$$

In this work, we utilize two specific types of side-information constraints (though a broader array is available; see, e.g., Sakos et al. (2023)). The first side-information constraint we use ensures that

the state space $\mathcal{X}$ of the system, i.e., the product of the simplices $\mathcal{X}_i \equiv \Delta(\mathcal{A}_i)$, $i = 1, \ldots, n$, is robust forward invariant with respect to the controlled dynamics in (2). This implies that, for any initialization $x(0) \in \mathcal{X}$, we have that $x(t) \in \mathcal{X}$ for all subsequent times $t > 0$, and for any control signal $\omega(t) \in \Omega$. To search over regressors that satisfy robust forward invariance (RFI), we rely on a specific characterization of the property that dictates a set remains robustly forward invariant under system (2) only if $f(x, \omega)$ lies in the tangent cone of $\mathcal{X}$ at $x$ for every control $\omega \in \Omega$. Using the characterization of the tangent cone at each simplex $\mathcal{X}_i$ (Nagumo, 1942), enforcing RFI is then reduced to verifying that, for all $x \in \mathcal{X}$ and $\omega \in \Omega$:

$$\sum_{a_i \in \mathcal{A}_i} p_{ia_i}(x, \omega) = 0,$$

$$p_{ia_i}(x, \omega) \geq 0, \quad \text{whenever } x_{ia_i} = 0. \tag{RFI}$$

The second type of side-information constraint we use is based on a fundamental assumption about agent behavior that arises from their strategic nature. The agents as strategic entities are expected to behave rationally, preferring actions that enhance their immediate benefits—a property known as positive correlation (PC) (Sandholm, 2010). Specifically, agents are inclined to choose actions that are likely to increase their expected utility, assuming other agents' behaviors remain unchanged, i.e., for all $x \in \mathcal{X}$ and $\omega \in \Omega$

$$\langle \nabla_{x_i} u_i(x, \omega), \, p_i(x, \omega) \rangle > 0, \text{ whenever } p(x, \omega) \neq 0. \tag{PC}$$

To enforce these side-information constraints computationally in our polynomial regression problem, we utilize sum-of-squares (SOS) optimization (Parrilo, 2000; Prestel & Delzell, 2001; Lasserre, 2001; Parrilo, 2003; Lasserre, 2006; Laurent, 2008). Both RFI and PC are represented as polynomial inequality or nonnegative constraints over the semialgebraic sets $\mathcal{X}$ and $\Omega$ (in PC's case, we need to relax the inequality (Sakos et al., 2023)). Using the SOS approach, instead of searching over polynomials $p(x)$ that are nonnegative over a semialgebraic set $\mathcal{S} \equiv \{x \mid g_j(x) \geq 0, h_\ell(x) = 0, j \in [m], \ell \in [r]\}$, we search over polynomials $p(x)$ that can be expressed as $p(x) = \sigma_0(x) + \sum_{j=1}^m \sigma_j(x)g_j(x) + \sum_{\ell=1}^r q_\ell(x)h_\ell(x)$ where $q_\ell$ are polynomials and $\sigma_j$ are sum-of-squares polynomials. Such polynomials $p$ are guaranteed to be nonnegative over $\mathcal{S}$, a condition that is also necessary under mild assumptions on the set $\mathcal{S}$ (Laurent, 2008, Theorem 3.20). Furthermore, for any given degree $d$, we can look for SOS certificates of degree $d$ through semidefinite programming, creating a hierarchy of semidefinite problems.

## 4.2 THE CONTROL STEP

After estimating the controlled dynamics which describes how players' strategies ($x$) respond to the incentives ($\omega$), our next goal is to steer the system towards desirable outcomes. To achieve this, we employ MPC, which formulates an optimization problem to identify the optimal sequence of control actions over a defined horizon, subject to constraints on control inputs. In our context, these control constraints represent practical limits on the incentives that can be offered to agents. The essence of MPC lies in its ability to leverage a mathematical model of the system to predict future behavior over a specified prediction horizon $T$. Each prediction is based on the current state measurement $x(t)$ and a sequence of future control signals $\omega_t := \{\omega_{0|t}, \ldots, \omega_{N-1|t}\} \subset \Omega$, which is calculated by solving a constrained optimization problem. Here, $N$ is the number of control steps within the control horizon $T := N \cdot \Delta t$ that determines the time period over which the control sequence is optimized. Typically, the objective function of the MPC is given by

$$J(\omega_t) = \sum_{n=0}^{N} \|x_{n|t} - x^*\|^2 + \alpha \sum_{n=0}^{N-1} \|\omega_{n|t}\|^2, \tag{4}$$

where $x_{n|t}$, $n = 1, \ldots, N$ correspond to values of the forecasted trajectory, and $x^*$ is a desirable target system state. The first term of $J(\omega_t)$ penalizes deviations of the predicted states $x_{n|t}$ from the target value $x^*$, while the latter term accounts for the control effort at weight $\alpha$. In addition to the above, large control signal variations can also be penalized by adding the term $\sum_{n=1}^{N} \|\omega_{n|t} - \omega_{n-1|t}\|^2$ at some desired weight. The optimal control sequence is obtained by solving the constrained optimiza-

tion problem

$$
\begin{aligned}
\min_{\omega_t} \quad & J(\omega_t) \\
\text{s.t.} \quad & \omega_{n|t} \in \Omega, \ 0 \le n \le N \\
& x_{n+1|t} = x_{n|t} + \Delta t \cdot p(x_{n|t}, \omega_{n|t}), \ 1 \le n \le N \\
& x_{0|t} = x(t).
\end{aligned}
\tag{5}
$$

The first control signal $\omega_{0|t}$ is then applied to the system and the optimization is repeated at time $t + \Delta t$ once the new state measurement $x(t + \Delta t)$ is obtained.

It is important to emphasize two key features of MPC: First, by optimizing control actions over a finite horizon, it can predict and maintain the system within predefined operational limits to avoid critical conditions. Second, by applying only the first control action at each time step, then shifting the horizon, and re-solving the optimization with the most up-to-date measurements, MPC can adapt to unexpected disturbances and generate a new sequence of control inputs accordingly.

## 5 EXPERIMENTS

In this section, we demonstrate the applicability and the performance improvements of the SIAR-MPC framework compared to combining MPC with solutions obtained from SINDYc and PINN, across various normal-form games of independent interest. The implementation of SINDYc follows the methodology described in Kaiser et al. (2018). For solving the sparse regression problem, we employ a sequential least squares procedure, as detailed in the supplementary information of Brunton et al. (2016). For PINN, the method is straightforward: side-information constraints are integrated into the neural network during its training as terms in the loss function that penalize violations of the desired constraints. In our examples, we enforce RFI and PC as such constraints. Given the limited number of training samples—in most cases, $5$ samples—we are restricted to a simple neural network architecture consisting of two hidden layers of size $5$. This limitation in the neural network's expressivity is counterbalanced by the simplicity of the ground-truth update policies $f$ (which, in most cases, are polynomial). As activation functions we use the $\tanh$ function. Finally, the side-information constraints are enforced using $2,500$ collocation points. Further details on the construction of the loss function and the generation of collocation points can be found in Appendix B.

We begin our demonstration with a well-known coordination game: the stag hunt. As coordination games correspond to non-cooperative setups where convergence to a socially optimal equilibrium is desirable, the stag hunt game is a natural candidate for comparing the performance of different system identification methods, combined with MPC, in guiding agents towards the game's optimal outcome. Next, moving away from the ideal landscape of coordination games, and into the more challenging regime of two-player zero-sum games, we demonstrate the performance of the SIAR-MPC framework in achieving two primary objectives: steering possibly non-polynomial learning dynamics with non-vanishing regret towards a Nash equilibrium of the game; and steering a provably chaotic system. In addition to the detailed analyses presented in this section, we conduct simulations across a large and diverse set of settings to gain a statistically significant understanding of each methods' performance. The results of these simulations can be found in Appendix A.

### 5.1 STAG HUNT GAME

The stag hunt game is a two-player two-action coordination game that models a strategic interaction in which both players benefit by coordinating their actions towards a specific superior outcome (the hunt of a stag). However, if that outcome is not possible, each player prefers to take advantage of the lack of coordination and come out on top of their opponent by choosing the alternative (hunt a rabbit by themselves) rather than coordinating to an inferior outcome (hunt a rabbit together). In this example, in the absence of control the players default to a stag hunt game given by the players' reward functions $u(\cdot) := u(\cdot, 0)$

$$
u_1(a) = u_2(a) = \mathbf{A}_{a_1, a_2}, \text{ where } \mathbf{A} = \begin{pmatrix} 4 & 1 \\ 3 & 3 \end{pmatrix}.
\tag{6}
$$

For expositional purposes, we restrict the game's evolution to a subset of symmetric two-player two-action games given by the control signals $\omega_{1,a_1,a_2}(t) = \omega_{2,a_2,a_1}(t) \in [0, 2]$ for all $t$. Furthermore,

for tractability purposes, we fix $\omega_{1,2,2}(t) := 0$. We assume that the agents' behavior is dictated by the replicator dynamics given, for player $i$ and action $a_i$ of $i$, by the polynomial update policies

$$f_{i,a_i}(x, \omega) = x_{i,a_i}(u_i(a_i, x_{-i}, \omega) - u_i(x, \omega)) \tag{7}$$

for all $x \in \mathcal{X}$ and $\omega \in \Omega$.

We are going to search for $f$ using the SIARc framework. Specifically, for each $i$ and $a_i$, we are going to search for polynomial $p_{i,a_i} : \mathcal{X} \times \Omega \to \mathbb{R}$ such that $p_{i,a_i}(x(t), \omega(t)) \approx f_{i,a_i}(x(t), \omega(t))$ for all $t \in \mathbb{R}_+$. As side-information constraints, we are going to impose the RFI and PC properties as given in the previous sections. Then, by substituting (6) to (PC) we have the following SIARc problem

$$
\begin{aligned}
\min_p \quad & \sum_{k=1}^{K} \left\| p\big(x(t_k), \omega(t_k)\big) - \dot{x}(t_k) \right\|^2 \\
\text{s.t.} \quad & p_{i,1}(x, \omega) + p_{i,2}(x, \omega) = 0, \ \forall i \\
& p_{i,1}\big((0,1), x_{-i}, \omega\big) \geq 0, \ \forall i \\
& p_{i,2}\big((1,0), x_{-i}, \omega\big) \geq 0, \ \forall i \\
& v_{i,1}(x, \omega)p_{i,1}(x, \omega) + v_{i,2}(x, \omega)p_{i,2} \geq 0, \ \forall i,
\end{aligned}
\tag{8}
$$

where $x \in \mathcal{X}$, $\omega \in \Omega$, and $v_{i,a_i} : \mathcal{X} \times \Omega \to \mathbb{R}$ are given by

$$v_{1,1}(x, \omega) = (4 + \omega_{1,1})x_{2,1} + (1 + \omega_{1,2})x_{2,2} \tag{9a}$$

$$v_{1,2}(x, \omega) = (3 + \omega_{2,1})x_{2,1} + 3x_{2,2} \tag{9b}$$

$$v_{2,1}(x, \omega) = (4 + \omega_{1,1})x_{1,1} + (1 + \omega_{1,2})x_{1,2} \tag{9c}$$

$$v_{2,2}(x, \omega) = (3 + \omega_{2,1})x_{1,1} + 3x_{1,2}. \tag{9d}$$

Since the update policies $f_{i,a_i}$ in (7) correspond to the replicator dynamics, the solution to the above optimization problem can be recovered by a 7-degree SOS relaxation (Sakos et al., 2023). Fig. 2 *(right)* shows the performance of the SIAR-MPC using this solution as a model for the MPC method. In the top panel of the figure, we have the trajectory $x(t)$ initialized at $x_0 = (0.4, 0.3)$ corresponding to the control signal $\omega(t)$ depicted in the bottom panel. The plot is divided into three sections, corresponding to the system identification phase, an evaluation period, and a control phase. In the first section, we set the control signals to normally distributed noise with mean zero (bounded in $\Omega$) and a sample of $K = 4$ datapoints from the resulting trajectory. In doing so, we achieve two things. On the one hand, the noise supplies variety between the data samples, which improves the efficiency of the system identification methods. On the other hand, we make sure to maintain a low aggregated control cost during the system identification phase. At the end of this phase, we solve the optimization problem in (8) and acquire a model of the system's update policies. In the second section of the plot, we compare the ground-truth dynamics with the model's predicted trajectory (in

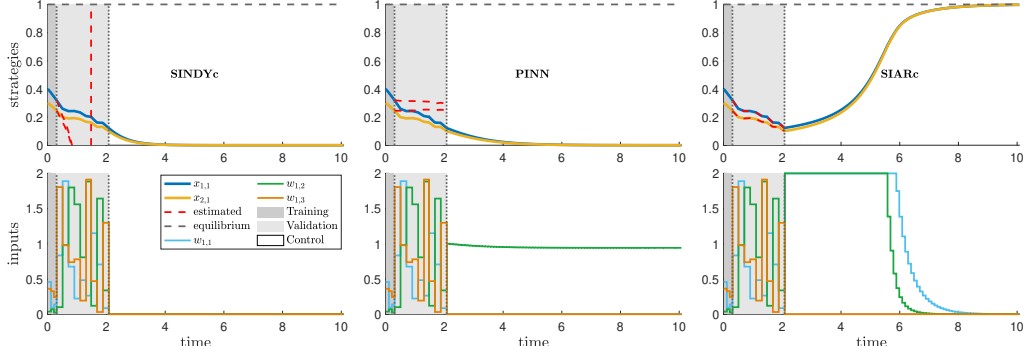

Figure 2: Performance comparison of the SINDY-MPC *(left)*, PINN-MPC *(center)*, and SIAR-MPC *(right)* solutions in steering the replicator dynamics for the stag hunt game. A dataset of 4 samples is used to train the system identification methods. Only the SIAR-MPC solution succeeds in steering the system to the superior Nash equilibrium in $x_{1,1}^* = x_{2,1}^* = 1$.

dashed red lines); here, the control signal is chosen randomly. Finally, in the steering phase, we steer the ground truth using the output of the MPC as the control signal: the objective is to steer the system to the superior Nash equilibrium of the stag hunt game at $x_{1,1}^* = x_{2,1}^* = 1$, which is achieved by the SIAR-MPC framework at $t \approx 8$. In comparison, the SINDY-MPC and PINN-MPC solutions (Fig. 2 *(left)* and Fig. 2 *(center)*, respectively) fail to complete the steering objective.

## 5.2 ZERO-SUM GAMES & CHAOS

The stag hunt game arguably provides an ideal landscape for the steering of game dynamics to a Nash equilibrium. This is because of the existence of a socially optimal Nash equilibrium (the hunt of a stag) that has a positive-measure basin of attraction. The class of zero-sum games, on the other hand, does not possess such merit, since in a zero-sum game, by definition, the player rewards sum up to zero independently of the strategies chosen by the players. To make matters worse, well-known game dynamics, e.g., the replicator dynamics, are known to exhibit undesirable behavior in zero-sum games such as Poincaré recurrence (Akin & Losert, 1984), or even chaos (Sato et al., 2002). Even if we amuse ourselves by instead pondering the more relaxed notion of time-average convergence, positive results in that area revolve around the notion of no-regret dynamics (Sorin, 2024), and the steering of learning dynamics of non-vanishing regret is, to the best of our knowledge, unexplored. In that regard, in the next couple of examples we demonstrate the performance of SIAR-MPC in steering the log-barrier dynamics—well-known learning dynamics of non-vanishing regret—in the matching pennies game and in the steering of chaotic replicator dynamics in an $\epsilon$-perturbed rock-paper-scissors ($\epsilon$-RPS) game. In both cases, we demonstrate the SIAR-MPC framework is able to steer the system towards the desired Nash equilibrium of the game.

**Matching Pennies Game**   The matching pennies game is a two-player, two-action zero-sum game where one player benefits by the existence of coordination among the two, while the other player benefits by the lack thereof. Formally, a matching pennies game is encoded in the uncontrolled players' reward functions in (1) by

$$u_1(a) = -u_2(a) = \mathbf{A}_{a_1, a_2}, \text{ where } \mathbf{A} = \begin{pmatrix} 1 & -1 \\ -1 & 1 \end{pmatrix}. \tag{10}$$

For similar reasons as in the previous example, we are going to restrict the game's evolution to a subset of two-player two-action zero-sum games given by $\omega_{1,a}(t) = \omega_{2,a}(t) \in [0,1]$ for all $t$, and set $\omega_{1,2,2}(t) := 0$. The log-barrier dynamics of the above time-varying game are, for player $i$ and action $a_i$ of $i$, given by the rational update policies

$$f_{i,a_i}(x, \omega) = x_{i,a_i}^2 \left( u_{i,a_i} - \frac{x_{i,1}^2 u_{i,1} + x_{i,2}^2 u_{i,2}}{x_{i,1}^2 + x_{i,2}^2} \right) \tag{11}$$

for all $t \in \mathbb{R}_+$ and $x \in \mathcal{X}$, where the shorthand $u_{i,x_j} := u_i(x_j, x_{-i}, \omega)$ is used for compactness. Observe that, as is the case for the updated policies of the replicator dynamics in (7), $f_i$ depends on $\omega(t)$ through $u_i$. In Fig. 3, we show that SIAR-MPC solution is able to steer a trajectory $x(t)$ of the

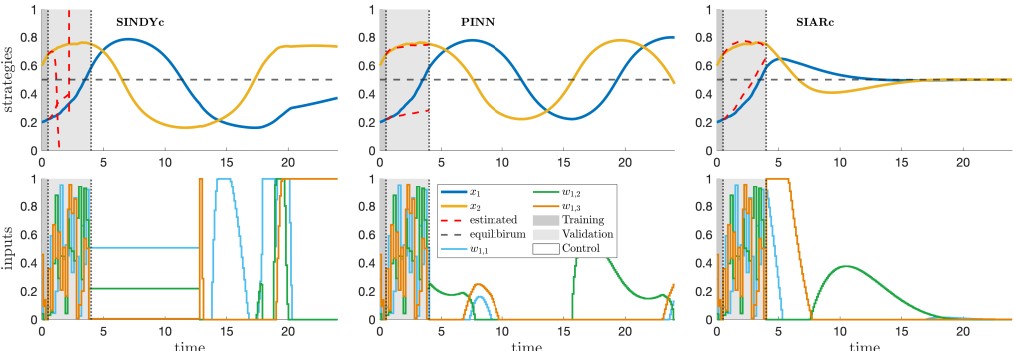

Figure 3: Performance comparison of the SINDY-MPC *(left)*, PINN-MPC *(center)*, and SIAR-MPC *(right)* solutions in steering the log-barrier dynamics for the matching pennies game.

above system initialized at $x(0) = (0.2, 0.6)$ to the unique mixed Nash equilibrium $x_{1,1}^* = x_{2,1}^* = 1/2$ of the matching pennies game with only $K = 6$ training samples.

**$\epsilon$-RPS Game** In our last example, we use the SIAR-MPC method to steer the replicator dynamics in an $\epsilon$-perturbed rock-paper-scissors game ($\epsilon$-RPS), a two-player, three-action zero-sum game where the replicator dynamics exhibit chaotic behavior (Sato et al., 2002; Hu et al., 2019). In a nutshell, this means that any two initialization of the system—even the ones that are infinitesimally close to each other—may lead to completely different trajectories. In other words, the accurate estimation of the agents' update policies is futile due to the finite precision of any numerical method. A $\epsilon$-RPS game may be encoded in the players' reward functions by replacing the payoff matrix in (10) with

$$\mathbf{A} = \begin{pmatrix} \epsilon & -1 & 1 \\ 1 & \epsilon & -1 \\ -1 & 1 & \epsilon \end{pmatrix}. \tag{12}$$

Due to the larger dimensionality of this game, compared to ones in the previous examples, we are going to consider time-evolving games in the subset of two-player three-action zero-sum games given by $\omega_{1,a}(t) = \omega_{2,a}(t) \in [-1, 1]$ for all $t$, and only four non-zero signals, namely, $\omega_{1,1,2}(t)$, $\omega_{1,1,3}(t)$, $\omega_{1,2,1}(t)$, and $\omega_{1,3,1}(t)$. In Fig. 4, we show the successful steering to the chaotic replicator dynamics for the $0.25$-RPS game, given as in equation 7, towards the unique mixed Nash equilibrium of the game $x_1^* = x_2^* = (1/3, 1/3, 1/3)$ based on a dataset of $K = 11$ samples.

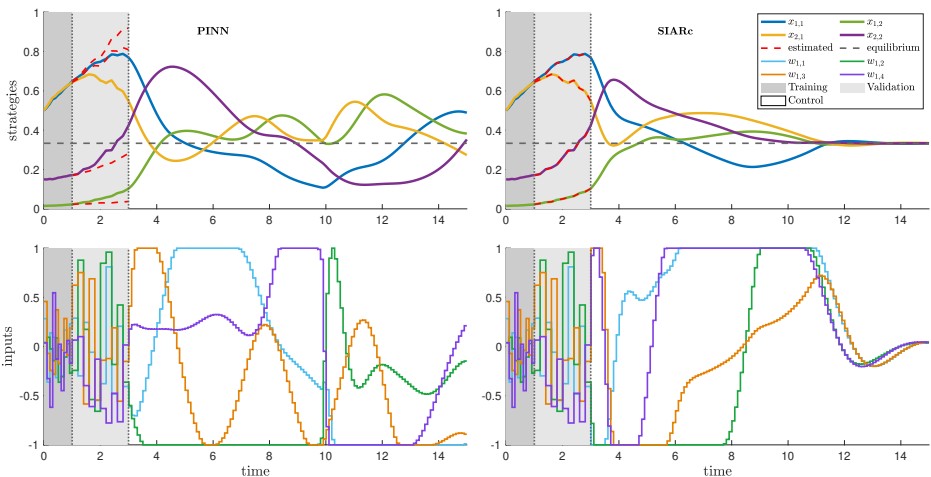

Figure 4: Performance comparison of the PINN-MPC, and SIAR-MPC solutions in steering the replicator dynamics for a $0.25$-RPS game.

## 6 CONCLUSIONS

In this work, we introduced SIAR-MPC, a new computational framework that extends SIAR for system identification of controlled game dynamics and integrates it with MPC for dynamic incentive adjustments, aiming to steer game dynamics towards desirable outcomes with limited data availability. Our results demonstrated that SIAR-MPC effectively steers systems towards optimal equilibria, stabilizes chaotic and cycling dynamics. Comparative analysis showed that SIAR-MPC outperforms alternative methods in data-scarce settings.

Future research can explore several potential directions. First, we intend to address a broader question: Given the inherent limitations of game dynamics, where convergence to NE is not always guaranteed, what are the necessary and sufficient conditions for achieving global stability in controlled game dynamics? Second, we aim to extend our framework to encompass games beyond the normal form, thereby expanding its applicability to a wider range of strategic interactions. Finally, we plan to investigate the scalability of our approach by exploring the uncoupling assumption, which could potentially enable its application to larger systems involving numerous agents.

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

## A    ADDITONAL EXPERIMENTAL RESULTS

To supplement the detailed analyses presented in Section 5, we conducted simulations across a wide range of diverse settings to obtain a statistically significant understanding of each methods' performance. For each setting, we generated 100 random initial conditions, and applied the SIAR-MPC, PINN-MPC and SINDY-MPC frameworks. The first three columns in the tables below present different metrics used to assess the performance of each framework, focusing on tracking accuracy, steady-state error and cost efficiency. Additionally, we generated 100 trajectories using SIARc, PINN and SINDYc to evaluate how accurately these models can replicate the true system dynamics.

### A.1    STAG HUNT GAME

We generated 100 initial conditions using Latin hypercube sampling and then applied SIAR-MPC, PINN-MPC and SINDY-MPC to the stag hunt game described in Section 5.1.

| Method | MSE (Ref.) | | Steady-State Error | | Cost | MSE (True) | |
|--------|------------|------------|--------------------|------------|------|------------|------------|
| | $x_{1,1}$ | $x_{2,1}$ | $x_{1,1}$ | $x_{2,1}$ | | $x_{1,1}$ | $x_{2,1}$ |
| SIARc | $3.41 \times 10^{-2}$ | $3.54 \times 10^{-2}$ | $2.26 \times 10^{-4}$ | $1.60 \times 10^{-4}$ | $5.02 \times 10^1$ | $1.21 \times 10^{-12}$ | $3.32 \times 10^{-9}$ |
| PINN | $5.65 \times 10^{-1}$ | $5.59 \times 10^{-1}$ | $6.55 \times 10^{-1}$ | $6.55 \times 10^{-1}$ | $1.09 \times 10^3$ | $7.37 \times 10^{-3}$ | $7.89 \times 10^{-3}$ |
| SINDYc | $6.67 \times 10^{-1}$ | $6.67 \times 10^{-1}$ | $7.80 \times 10^{-1}$ | $7.80 \times 10^{-1}$ | $1.27 \times 10^3$ | $1.05 \times 10^3$ | $6.50 \times 10^2$ |

Table 1: Results for the stag hunt game across 100 initial conditions. The first three metrics compare the performance of the SIAR-MPC, PINN-MPC and SINDY-MPC frameworks: **MSE(Ref.)** shows the mean squared error between the estimated and the reference trajectories; **Steady-State Error** measures the deviation of the final state from the reference state; **Cost** represents the accumulated cost of steering the system to the reference state. The last metric, **MSE(True)**, evaluates the accuracy of the SIARc, PINN, and SINDYc models in predicting the true system dynamics by reflecting the average squared error between the estimated $\dot{x}$ and the true $\dot{x}$. All results are averaged across the initial conditions. The results clearly demonstrate that SIAR-MPC and SIARc consistently achieve lower error values and control cost compared to the other methods.

### A.2    MATCHING PENNIES GAME

We generated 100 initial conditions using Latin hypercube sampling. We then applied SIAR-MPC, PINN-MPC and SINDY-MPC to the matching pennies game described in Section 5.2. However, as the latter two methods are data-driven techniques, for fairness we did the comparison based on a larger training dataset of $K = 50$ samples.

| Method | MSE (Ref.) | | Steady-State Error | | Cost | MSE (True) | |
|--------|------------|------------|--------------------|------------|------|------------|------------|
| | $x_{1,1}$ | $x_{2,1}$ | $x_{1,1}$ | $x_{2,1}$ | | $x_{1,1}$ | $x_{2,1}$ |
| SIARc | $5.13 \times 10^{-2}$ | $6.30 \times 10^{-2}$ | $9.90 \times 10^{-2}$ | $1.00 \times 10^{-1}$ | $2.26 \times 10^2$ | $6.25 \times 10^{-4}$ | $1.32 \times 10^{-3}$ |
| PINN | $7.56 \times 10^{-2}$ | $7.83 \times 10^{-2}$ | $2.13 \times 10^{-1}$ | $1.88 \times 10^{-1}$ | $2.69 \times 10^2$ | $6.50 \times 10^{-3}$ | $6.26 \times 10^{-3}$ |
| SINDYc | $9.62 \times 10^{-2}$ | $1.05 \times 10^{-1}$ | $2.71 \times 10^{-1}$ | $2.90 \times 10^{-1}$ | $3.00 \times 10^9$ | $7.97 \times 10^2$ | $2.31 \times 10^5$ |

Table 2: Results for the matching pennies game across 100 initial conditions. We compare the performance of the SIARc, PINN and SINDYc models across the metrics described in Table 1. The results clearly demonstrate that SIARc consistently achieve lower error values and control cost compared to the other methods.

### A.3    $\epsilon$-RPS GAME

We generated 100 initial conditions by uniformly sampling from the 3-dimensional simplex and then applied SIAR-MPC, PINN-MPC and SINDY-MPC to the $\epsilon$-RPS game described in Section 5.2.

| Method | MSE (Ref.) | | | | Steady-State Error | | | | Cost |
|--------|-----------|--------|--------|--------|-----------|--------|--------|--------|------|
| | $x_{1,1}$ | $x_{1,2}$ | $x_{2,1}$ | $x_{2,2}$ | $x_{1,1}$ | $x_{1,2}$ | $x_{2,1}$ | $x_{2,2}$ | |
| SIARc | $9.73 \times 10^{-3}$ | $1.09 \times 10^{-2}$ | $1.01 \times 10^{-2}$ | $7.90 \times 10^{-3}$ | $3.03 \times 10^{-3}$ | $2.14 \times 10^{-3}$ | $3.14 \times 10^{-3}$ | $4.92 \times 10^{-3}$ | $2.68 \times 10^{1}$ |
| PINN | $3.50 \times 10^{-2}$ | $3.65 \times 10^{-2}$ | $1.71 \times 10^{-2}$ | $2.89 \times 10^{-2}$ | $1.06 \times 10^{-1}$ | $1.44 \times 10^{-1}$ | $5.98 \times 10^{-2}$ | $7.45 \times 10^{-2}$ | $1.03 \times 10^{2}$ |
| SINDYc | $7.14 \times 10^{-2}$ | $5.29 \times 10^{-2}$ | $5.54 \times 10^{-2}$ | $5.11 \times 10^{-2}$ | $2.39 \times 10^{-1}$ | $2.10 \times 10^{-1}$ | $1.92 \times 10^{-1}$ | $2.06 \times 10^{-1}$ | $7.04 \times 10^{34}$ |

Table 3: Results for the $\epsilon$-RPS game for 100 initial conditions. We compare the performance of the SIARc, PINN and SINDYc models across the metrics MSE (Ref.), Steady-State Error, and Cost as described in Table 1. The results clearly demonstrate that SIARc consistently achieve lower error values and control cost compared to the other methods.

| Method | MSE (True) | | | |
|--------|-----------|--------|--------|--------|
| | $x_{1,1}$ | $x_{1,2}$ | $x_{2,1}$ | $x_{2,2}$ |
| SIARc | $1.18 \times 10^{-7}$ | $2.48 \times 10^{-7}$ | $1.02 \times 10^{-9}$ | $3.08 \times 10^{-10}$ |
| PINN | $1.31 \times 10^{-2}$ | $1.25 \times 10^{-2}$ | $9.05 \times 10^{-3}$ | $1.25 \times 10^{-2}$ |
| SINDYc | $7.48 \times 10^{7}$ | $2.92 \times 10^{4}$ | $1.42 \times 10^{8}$ | $1.95 \times 10^{6}$ |

Table 4: Results for the $\epsilon$-RPS game for 100 initial conditions. We compare the performance of the SIARc, PINN and SINDYc models in terms of MSE (True) as described in Table 1. The results clearly demonstrate that SIARc consistently achieve lower error values compared to the other methods.

## A.4    STATE AVOIDANCE CONSTRAINTS

In this section, we revisit the example of a matching pennies game to illustrate a key feature of MPC: its ability to incorporate state constraints (Camacho & Bordons, 2007). In our case, the purpose of these state constraints is to keep the system trajectory away from specific, undesirable areas of the state space. Consider a scenario where the policymaker aims to steer the system towards the mixed Nash equilibrium at $x_{1,1}^* = x_{2,1}^* = 1/2$, while simultaneously ensuring that the system trajectory avoids a ball of radius $0.4$ centered at $[(0,1),(1,0)]$. This constraint can be incorporated into the MPC problem and as illustrated in Fig. 5, SIAR-MPC can steer the system to the desired equilibrium while also avoiding the restricted area of the state space (shaded in blue).

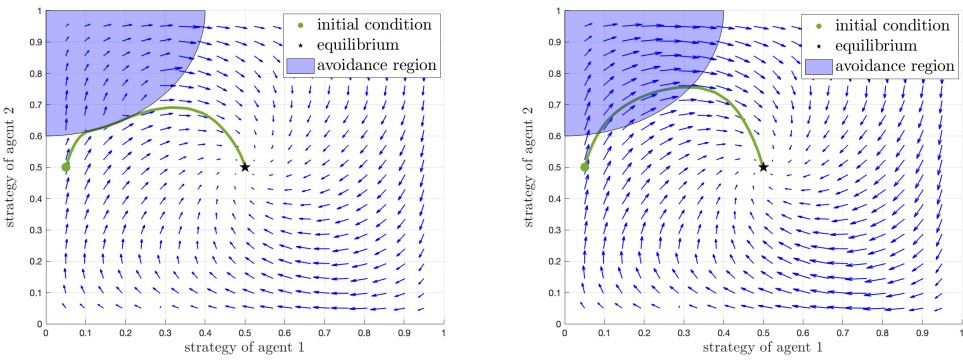

Figure 5: Vector field of the steering direction of the SIAR-MPC when the additional state avoidance constraint is *(left)*, and is not *(right)* imposed. The restricted area is shaded in blue. The green line corresponds to a controlled trajectory of the replicator dynamics for the matching pennies game.

## B DETAILS OF PINN

Consider a two-player two-action game. Given that each player has two actions, the state space can be reduced to $(x_1, x_2) \in [0, 1] \times [0, 1]$, where $x_1$ and $x_2$ represent the first state of each player[1]. We train a neural network to approximate the dynamics $f_1$ and $f_2$ such that

$$\dot{x}_i = f_i(x_1, x_2, \omega) \approx p_i(x_1, x_2, \omega) = \dot{\hat{x}}_i, \ i = 1, 2,$$

where $\omega \in \Omega$. The neural network takes $x_1, x_2, \omega$ as inputs and outputs an approximation $\dot{\hat{x}}_i$ of $\dot{x}_i$. The training dataset is generated as described in Section 4.1 and is represented as $D := \{(x_{1,i}, x_{2,i}, \omega_i, \dot{x}_{1,i}, \dot{x}_{2,i})\}_{i=1}^{N_D}$.

### B.1 PHYSICS INFORMATION AND COLLOCATION DATASET GENERATION

To incorporate the physical knowledge into the PINN training, we consider two sets of side information constraints: Robust Forward Invariance (RFI) and Positive Correlation (PC).

*Robust Forward Invariance*

For the reduced state space, the RFI constraint can be written as:

$$p_1(0, x_2, \omega) \geq 0, \quad \forall x_2 \in [0, 1], \ \omega \in \Omega$$
$$p_1(1, x_2, \omega) \leq 0, \quad \forall x_2 \in [0, 1], \ \omega \in \Omega$$
$$p_2(x_1, 0, \omega) \geq 0, \quad \forall x_1 \in [0, 1], \ \omega \in \Omega$$
$$p_2(x_1, 1, \omega) \leq 0, \quad \forall x_1 \in [0, 1], \ \omega \in \Omega$$

To enforce this side information constraint during training, we generate a set of $N_C^{\text{RFI}}$ collocation points. Specifically, we create the following sets:

$$C_1^{\text{RFI}} := \{(x_{1,i}, x_{2,i}, \omega_i)\}_{i=1}^{N_{C_1}^{\text{RFI}}} \text{ with } x_{1,i} = 0, \ x_{2,i} \sim U[0, 1] \text{ and } \omega_i \sim U\Omega$$

$$C_2^{\text{RFI}} := \{(x_{1,i}, x_{2,i}, \omega_i)\}_{i=1}^{N_{C_2}^{\text{RFI}}} \text{ with } x_{1,i} = 1, \ x_{2,i} \sim U[0, 1] \text{ and } \omega_i \sim U\Omega$$

$$C_3^{\text{RFI}} := \{(x_{1,i}, x_{2,i}, \omega_i)\}_{i=1}^{N_{C_3}^{\text{RFI}}} \text{ with } x_{2,i} = 0, \ x_{1,i} \sim U[0, 1] \text{ and } \omega_i \sim U\Omega$$

$$C_4^{\text{RFI}} := \{(x_{1,i}, x_{2,i}, \omega_i)\}_{i=1}^{N_{C_4}^{\text{RFI}}} \text{ with } x_{2,i} = 1, \ x_{1,i} \sim U[0, 1] \text{ and } \omega_i \sim U\Omega$$

where $x \sim U\mathcal{S}$ denotes that $x$ is sampled uniformly from set $\mathcal{S}$. The combined set of collocation points for RFI is then $C^{\text{RFI}} = C_1^{\text{RFI}} \cup C_2^{\text{RFI}} \cup C_3^{\text{RFI}} \cup C_4^{\text{RFI}}$.

*Positive Correlation*

For the reduced state space, the PC constraints can be written as:

$$\langle \nabla_{x_1} u_1(x_1, x_2, \omega), \ p_1(x_1, x_2, \omega) \rangle \geq 0, \forall x_1, x_2 \in [0, 1], \ \omega \in \Omega$$
$$\langle \nabla_{x_2} u_2(x_1, x_2, , \omega), \ p_2(x_1, x_2, , \omega) \rangle \geq 0, \forall x_1, x_2 \in [0, 1], \ \omega \in \Omega$$

To enforce this side information constraint during training, we generate a set of $N_C^{\text{PC}}$ collocation points. Specifically, we create the following set:

$$C^{\text{PC}} := \{(x_{1,i}, x_{2,i}, \omega_i)\}_{i=1}^{N_C^{\text{PC}}} \text{ with } x_{1,i}, \ x_{2,i} \sim U[0, 1] \text{ and } \omega_i \sim U\Omega.$$

### B.2 PINN LOSS FUNCTION

The loss function for PINN integrates three components: a loss function for the supervised learning over dataset $D$ and two physics-informed loss functions over the collocation points $C^{\text{RFI}}$ and $C^{\text{PC}}$ to enforce side information constraints RFI and PC.

---

[1]Note that the second state of each player is simply $1 - x_i$ for $i = 1, 2$, and thus, the corresponding dynamics are given by $-f_i$ for each $i = 1, 2$.

*Supervised Loss*

The supervised loss is the standard squared error loss to ensure neural netwrok predictions align closely with the training data and is given as:

$$\ell^0 = \sum_{(x_1, x_2, \omega, \dot{x}_1, \dot{x}_2) \in D} (p_1(x_1, x_2, \omega) - \dot{x}_1)^2 + (p_2(x_1, x_2, \omega) - \dot{x}_2)^2$$

*Physics Loss*

The physics loss for RFI is defined as:

$$\ell^{\text{RFI}} = \sum_{(x_1, x_2, \omega) \in C_1^{\text{RFI}}} \max(0, -p_1(x_1, x_2, \omega)) + \sum_{(x_1, x_2, \omega) \in C_2^{\text{RFI}}} \max(0, p_1(x_1, x_2, \omega))$$

$$+ \sum_{(x_1, x_2, \omega) \in C_3^{\text{RFI}}} \max(0, -p_2(x_1, x_2, \omega)) + \sum_{(x_1, x_2, \omega) \in C_4^{\text{RFI}}} \max(0, p_2(x_1, x_2, \omega)).$$

Similarly, the physics loss for the PC constraint is defined as:

$$\ell^{\text{PC}} = \sum_{(x_1, x_2, \omega) \in C^{\text{PC}}} \max\left(0, -\nabla_{x_1} u_1(x_1, x_2, \omega) \cdot p_1(x_1, x_2, \omega)\right)$$

$$+ \sum_{(x_1, x_2, \omega) \in C^{\text{PC}}} \max\left(0, -\nabla_{x_2} u_2(x_1, x_2, \omega) \cdot p_2(x_1, x_2, \omega)\right).$$

Then, the loss function for PINN with a weighted summation is given as:

$$\ell = \lambda_0 \ell^0 + \lambda_{\text{RFI}} \ell^{\text{RFI}} + \lambda_{\text{PC}} \ell^{\text{PC}}.$$

