# OpenReview forum: "Learning and Steering Game Dynamics Towards Desirable Outcomes"
_ICLR.cc/2025/Conference — ICLR 2025 Conference Withdrawn Submission_

### Official Review · Reviewer_1684 · 2024-10-30

**Soundness:** 2
**Presentation:** 3
**Contribution:** 2
**Rating:** 3
**Confidence:** 4

**Summary:**

This paper proposes a new framework, SIAR-MPC, to address undesirable behaviors in game dynamics. The framework consists of two steps: first, it identifies controlled system dynamics using a polynomial regressor, incorporating side information as additional constraints to improve the accuracy of the learned dynamics. Second, Model Predictive Control (MPC) is adapted to predict the desired control actions.

**Strengths:**

1.	The paper presents an interesting setting with a clear motivation, making it easy to follow and understand. The approach of first identifying system dynamics and then planning to control the system is particularly intriguing.

2.	The use of polynomial regression with side-information constraints (RFI and PC) and the application of sum-of-squares (SOS) optimization shows a solid foundation in mathematical. The framework also leverages MPC effectively to solve constrained optimization problems dynamically.

**Weaknesses:**

1.	Some key concepts in the text lack clear definitions or explanations, which may confuse readers; further clarification is recommended. For example, the concept of "side-information constraints"(First shown at Page 2 Lines 56) is central to the paper, yet it lacks a clear definition and explanation. It’s not evident what constitutes side information and how it contributes to enhancing the accuracy of the learned controlled dynamics. The term "Strategic Nature" (Page 5, Line 230) is mentioned to justify the validity of the second side-information constraint. However, what actually plays a crucial role in supporting this constraint is the concept of Positive Correlation (PC). It's unclear why the authors introduced the notion of "Strategic Nature" in this context.

2.	Lack of theoretical justification. The paper does not provide evidence that a polynomial regressor is sufficient to accurately capture system dynamics, especially given the limited number of samples (K=5). The two side-information constraints are proposed to aid in learning an accurate model of the controlled system dynamics with limited data. However, there is no theoretical justification provided on how these constraints contribute to this goal. This is particularly concerning given that the second step involves MPC, which requires a high-fidelity model. Additionally, the use of SOS optimization introduces further uncertainty in achieving a precise model.

3.	Experimental issues: In the first paragraph of Experiments (Page 6, lines 294), the neural network consisting of two hidden layers of size 5 is trained with only 5 samples, which arose the problem of underfitting. The maximum number of samples used in the training phase is 11, for such a scare data, the comparison between any neural network-based method with the proposed method is unfair. Additionally, the baselines (PINNs from 2019 and SINDYc from 2018) are relatively outdated. More recent methods, such as Phycrnet, are mentioned in the related work. Besides, in data-scarce settings, traditional linear programming methods like pseudospectral method, optimal control(based on the Pontryagin maximum principle) should be considered.

**Questions:**

1.	Why did the authors choose these two specific side-information constraints among all possible options as listed in the reference?

2.	In every experiment, there is only one initial reward matrix. Can the proposed method achieve similar performance with different reward values?

3.	How critical is MPC in this approach? How does the prediction horizon impact performance? It would be helpful if the authors could provide additional experiments to explore this.

---

> ### Author Response · Authors · 2024-11-23
>
> We are grateful to the reviewer for their insightful feedback. We address the reviewer's concerns below.
>
> **1. Some key concepts in the text lack clear definitions or explanations, which may confuse readers; further clarification is recommended. For example, the concept of "side-information constraints"(First shown at Page 2 Lines 56) is central to the paper, yet it lacks a clear definition and explanation. It’s not evident what constitutes side information and how it contributes to enhancing the accuracy of the learned controlled dynamics. The term "Strategic Nature" (Page 5, Line 230) is mentioned to justify the validity of the second side-information constraint. However, what actually plays a crucial role in supporting this constraint is the concept of Positive Correlation (PC). It's unclear why the authors introduced the notion of "Strategic Nature" in this context.**
>
> We will address your concern by providing a clearer definition of "side information" in the relevant sections of the revised manuscript. Side information refers to additional knowledge about the learning dynamics of agents or assumptions regarding their behavior beyond the trajectory data. Specifically, in our setting, it is any knowledge that can be expressed as polynomial non-negativity or equality constraints over a semi-algebraic set.
>
> "Positive correlation" refers to the positive correlation between the directions of motion and a quantity known in the literature as the pseudogradient of the players' payoff. This relationship expresses the rationality of the players, as it encodes that players prefer directions that do not decrease their expected payoff. This is clearly an information about the strategic nature of the players.
>
> **2. Lack of theoretical justification. The paper does not provide evidence that a polynomial regressor is sufficient to accurately capture system dynamics, especially given the limited number of samples (K=5). The two side-information constraints are proposed to aid in learning an accurate model of the controlled system dynamics with limited data. However, there is no theoretical justification provided on how these constraints contribute to this goal. This is particularly concerning given that the second step involves MPC, which requires a high-fidelity model. Additionally, the use of SOS optimization introduces further uncertainty in achieving a precise model.**
>
> The SIAR solution is a regressor, and therefore its quality depends on the quality and number of samples. The benefit of SIAR is that it incorporates side-information, and thus reducing the size of the problem's feasible set. To the best of our knowledge, there are no theoretical results that quantify the improvements such a reduction provides to the regressor. A relevant question is whether, a polynomial regressor, as returned by SIAR, can be inherently "bad". In that regard, the results of [R1] can readily be extended in the presence of control, and therefore the existence of a "good" polynomial regressor (a feasible solution) that satisfies the given side-information constraints with SOS certificates is guaranteed to exist for some level of the SOS hierarchy. We will add the proof in the appendix of the revised manuscript.
>
> [R1] Sakos, Iosif, Antonios Varvitsiotis, and Georgios Piliouras. "Data-Scarce Identification of Game Dynamics via Sum-of-Squares Optimization" arXiv preprint arXiv:2307.06640 (2023).

---

> > ### Author Response · Authors · 2024-11-23
> >
> > **3. Experimental issues: In the first paragraph of Experiments (Page 6, lines 294), the neural network consisting of two hidden layers of size 5 is trained with only 5 samples, which arose the problem of underfitting. The maximum number of samples used in the training phase is 11, for such a scare data, the comparison between any neural network-based method with the proposed method is unfair. Additionally, the baselines (PINNs from 2019 and SINDYc from 2018) are relatively outdated. More recent methods, such as Phycrnet, are mentioned in the related work. Besides, in data-scarce settings, traditional linear programming methods like pseudospectral method, optimal control(based on the Pontryagin maximum principle) should be considered.**
> >
> > Our primary motivation is to demonstrate how incorporating additional system knowledge can compensate for limited data, improving both performance and accuracy. For benchmarking, we specifically sought approaches that align with this principle. We understand that one can find other methods that fit into this category like the ones mentioned by the reviewer. Our aim was to compare our proposed method with a foundational and widely recognized approach in this area, which PINNs represent. PINNs were chosen because they incorporate extra knowledge of the system into the neural network training process, making them particularly effective in data-scarce settings compared to standard neural networks. Moreover, to the best of our knowledge, none of the methods mentioned by the reviewer have been applied to the extreme data-scarce regime we are addressing.
> >
> > Regarding the fairness of the comparison, our experiments are designed to highlight how leveraging system knowledge (as both PINNs and our approach do) can improve performance in data-scarce settings. While the number of training samples is indeed limited, this limitation is intentional and reflects the motivation for our study—developing methods that can succeed when data is scarce and system knowledge is leveraged. Additionally, it is important to note that the loss function for PINNs comprises three components: the standard supervised learning loss over the training data and two additional physics-informed loss functions over the collocation points to enforce side information constraints. This means that PINNs benefit from a larger effective dataset due to the inclusion of the collocation points, which, in our case, amounts to 2000 points. Thus, the data for training PINNs is not as limited as it might initially appear.
> >
> > **4. Why did the authors choose these two specific side-information constraints among all possible options as listed in the reference?**
> >
> > We chose these two specific side-information constraints because they are among the most general and interpretable options, making them easier to integrate and present within the context of our work. Specifically, forward invariance is something that needs to hold of dynamics in a state space, and positive correlation is one of the most natural examples of strategic information that can be incorporated. This choice of two side information constraints ensures clarity and accessibility for readers. Additionally, interested readers can refer to the reference given and incorporate other forms of side information as needed for their specific problems. For instance, if it is known that the game is a congestion game, one could assume anonymity and use that as an additional side-information constraint.

---

> > > ### Author Response · Authors · 2024-11-23
> > >
> > > **5. In every experiment, there is only one initial reward matrix. Can the proposed method achieve similar performance with different reward values?**
> > >
> > > To address this question, we conducted additional simulations to demonstrate that the proposed SIAR-MPC method achieves consistent performance even when the payoff matrix is altered. Specifically, for the Stag Hunt Game, we generated 50 random payoff matrices while ensuring that the game maintained its cooperative nature with a socially optimal equilibrium. We evaluated the performance of SIAR-MPC across 50 systems and compared it with other methods based on our performance metrics, as detailed in Appendix A of the manuscript.
> > >
> > > |  Method  |       MSE (Ref.) x₁,₁       |       MSE (Ref.) x₂,₁       |     SS Error x₁,₁      |     SS Error x₂,₁      |       Cost       |
> > > |:--------:|:---------------------------:|:---------------------------:|:---------------------:|:---------------------:|:----------------:|
> > > |  SIARc   |      2.01 × 10⁻¹           |      1.80 × 10⁻¹           |      1.80 × 10⁻¹      |      1.80 × 10⁻¹      |    3.71 × 10²    |
> > > |   PINN   |      5.78 × 10⁻¹           |      5.47 × 10⁻¹           |      5.80 × 10⁻¹      |      5.80 × 10⁻¹      |    1.08 × 10³    |
> > > |  SINDYc  |      5.38 × 10⁻¹           |      4.87 × 10⁻¹           |      5.52 × 10⁻¹      |      5.36 × 10⁻¹      |    1.70 × 10⁷    |
> > >
> > > The results given in the above table clearly demonstrates that SIAR-MPC consistently achieves lower error values and control cost compared to the other methods.
> > >
> > > **6. How critical is MPC in this approach? How does the prediction horizon impact performance? It would be helpful if the authors could provide additional experiments to explore this.**
> > >
> > > We use MPC primarily because it is a well-established optimal control method, offering several key advantages for our approach. First, it provides the flexibility to correct and adapt as we proceed, which is particularly important since we do not assume the true model is known. Most importantly, MPC is well-suited for handling constraints, which is highly relevant in our setting as there are practical limits to the incentives that can be offered to players.
> > >
> > > The prediction horizon significantly impacts performance by introducing a trade-off between computational efficiency and MPC's ability to steer the system effectively within the given horizon. A very short prediction horizon increases the likelihood that MPC will fail to compute a suitable control input, as the target state may not be achievable within such a limited timeframe. On the other hand, a very long prediction horizon can lead to significantly increased computational costs, potentially making real-time implementation impractical. Therefore, selecting an appropriate prediction horizon is a design choice that balances computational feasibility with the need to achieve the desired system outcomes effectively.
> > >
> > > We also welcome any suggestions for alternative control methods and would be happy to test them in our experiments.

---

### Official Review · Reviewer_iVZL · 2024-11-04

**Soundness:** 3
**Presentation:** 4
**Contribution:** 2
**Rating:** 5
**Confidence:** 3

**Summary:**

This paper investigates the problem of steering agent behaviors in normal-form games.There is a central planner being able to influence the game's utility function. The agents change their policy according to the current state of the game. The paper proposes the SIRC-MPC framework. In this framework the planner first learn the agent's behavior by fitting the dynamics with polynomial regressors. To facilitate the learning, RFI and PC are incorporated as regularizations. Then it steers the behavior via a MPC. Finally it conducts experiments to illustrate the effectiveness of this framework.

**Strengths:**

1. The paper is clearly written and easy to follow.
2. This framework is very general. Only reasonable constraints are placed to enhance the sample efficiency of the learning.

**Weaknesses:**

1. The technical novelties of this paper is a bit unclear to me. See questions below.
2. The motivation of this paper is a bit unclear to me. See questions below.

**Questions:**

1. Can the authors clarify on the technical novelties of this paper? For example, one contribution of this paper is its superiority in data-scarce settings. However using PINN enhances performance in this setting is straightforward to me. Is there some technical difficulty I am missing here?
2. For the central planner, steering is not free. Larger $\omega$ is clearly more costly in real-world application. Should we compare the algorithms under the fixed budget?
3. This paper places no constraint on how we choose $\omega$ in the first phase. On the one hand, we cannot intervene a real-world game arbitrarily at our wishes, so this seems to be a strong constraint. On the other hand, if we allow online learning, i.e. adaptively picking $\omega$ so that the data is more informative, the sample complexity should be even lower. Is online learning a more natural setting?

---

> ### Author Response · Authors · 2024-11-23
>
> We are grateful to the reviewer for their insightful feedback. We address the reviewer's concerns below.
>
> **1. Can the authors clarify on the technical novelties of this paper? For example, one contribution of this paper is its superiority in data-scarce settings. However using PINN enhances performance in this setting is straightforward to me. Is there some technical difficulty I am missing here?**
>
> We are not sure what the reviewer means by saying that ``using PINN enhances performance in this setting is straightforward'', though. Could the reviewer please clarify this point?
>
> Our contributions lie in demonstrating that SIARc can perform well in scenarios where PINNs and SINDYc do not do well. In particular, our examples show settings where PINN is not able to converge to the target equilibrium, while SIARc consistently achieves this. Additionally, we conducted further experiments, detailed in the Appendix A, where SIARc outperforms PINN across multiple metrics, further highlighting its robustness in extreme data-scarce settings.
>
> **2. For the central planner, steering is not free. Larger $\omega$ is clearly more costly in real-world application. Should we compare the algorithms under the fixed budget?**
>
> For clarity, in this work we assume that the budget is fixed per time. The accumulated budget is not constrained. This is by design in order to include games that target Nash equilibrium is unstable, e.g., any non-zero-sum game with a unique interior Nash equilibrium. As a well-known example, we refer the reviewer to [R3]. We are going to add such an example to further motivate this idea.
>
> **3. This paper places no constraint on how we choose $\omega$ in the first phase. On the one hand, we cannot intervene a real-world game arbitrarily at our wishes, so this seems to be a strong constraint. On the other hand, if we allow online learning, i.e. adaptively picking so that the data is more informative, the sample complexity should be even lower. Is online learning a more natural setting?**
>
> In our framework, incentives are conceptualized as control inputs designed by the MPC that affect the payoff matrix. As the payoff matrix, by definition, encodes all the information about the player's incentives, altering the payoffs is the natural  method for influencing agent behavior. Which payoffs can be affected is a setting-specific question: for example, in our RPS game we only allow the controller to add incentives to certain outcomes, and these restrictions hold for all phases of our framework (training, verification, and control).
>
> In all phases of our framework, we also impose lower and upper bounds on the control input $\omega$, effectively enforcing a fixed budget per time step. Additionally, for the training phase, we scale the control input and explain the rationale behind this decision  (see Section 5.1, line 355). Specifically, $\omega$ is sampled from a normal distribution with a mean of zero, constrained within the predefined upper and lower limits. This ensures that the aggregated control cost remains low during the system identification phase.
>
> Randomization of the control inputs in the training phase is an easy way to get different control inputs for learning how the dynamics depends on the control, and we show empirically that it suffices for learning in our examples. We do not think the method of choosing control inputs in this phase matters much but agree with the reviewer that there might be more natural ways of selecting control inputs rather than randomly generating them, and that could be the subject of future work.
>
> We do not see how an online learning method would work better though. Could the reviewer clarify the intuition behind this?
>
> [R3] Robert Kleinberg, Katrina Ligett, Georgios Piliouras and Éva Tardos, "Beyond the Nash Equilibrium Barrier," In 2010 Innovations in Computer Science (ICS), pp. 125-140, 2010.

---

> > ### Comment · Reviewer_iVZL · 2024-11-24
> >
> > 1. Let me rephrase my first question. This paper is well-motivated and the framework is well-established, but the design of the framework or the experiment does not let me learn anything new. As an example, I would not be surprised if I am told that PINN can improve the SIARc framework in data-scarce setting. This is because PINN has more inductive bias than a general model, so it should be more sample efficient. First identifying the system and then using control algorithm also seems to be the algorithm one would first try. I would be happier if I am presented with i) theorem(s) stating the framework can steer some class of games within some budget to the highest social welfare, ii) realistic experiment rather than didactic examples showing the framework can be used to mitigate social dilemmas (maybe a large-scale traffic problem), or iii) interesting insight regarding how should one steer the dynamics or what should one avoid. I might be missing something but unfortunately I do not see any of them yet. Hence I want the author to clarify what is the main novelty of this paper, what is the technical difficulty of this paper or why should I be surprised by the result.
> > 2. To my understanding, not all $\omega$s in the system identification phase are equally important. This is because one only cares about the agents' behavior that exhibits under the final optimal control sequence. So a rough idea is we first explore the system by uniformly randomly sampling, get a rough estimate of what our control sequence should be like, and then concentrate the sampling in regions that are close to the desired sequence. This is like the Explore-then-Commit vs Upper-Confidence-Bound algorithm in Multi-Armed Bandits. This seems to be a natural improvement to the proposed framework. Is there any reason this is not feasible in the setting this paper is considering?

---

> > > ### Author Response · Authors · 2024-12-02
> > >
> > > Thank you for your comments and questions. We appreciate the interest shown in our paper.
> > >
> > > 1. Thank you for your concerns and suggestions: we do agree that any of the three things you raised would improve our manuscript. Regarding theory, we do have a theorem about the ability of polynomial dynamics to approximate any continuously differentiable dynamics even in the control setting which we will include in an updated version of the manuscript. This provides theoretical justification for using SIARc to learn a good model of the dynamics during the learning phase. However, theoretical guarantees for control, and in particular MPC, in the setting of nonlinear dynamics are notoriously difficult to achieve (beyond statements like local stability under multiple assumptions), and we are thus not able to have theoretical guarantees for the whole framework. Our basis for using the framework lies in SIAR’s theory as well as empirical evidence to be able to learn from few data, together with MPC’s continual sampling and optimization to be able to overcome small inaccuracies in the learnt model, and is backed up by our experiments.
> > >
> > > We agree that more realistic experiments would showcase the efficacy of the method more convincingly. That, together with methods for scaling up the framework, shall be the subject of future work. Using the results of SIAR+MPC to develop heuristic insights on control inputs for steering can also be the subject of future work, but is beyond the scope of our current paper.
> > >
> > > The purpose of our paper was to bring previous techniques, which both have convincing empirical efficacy in their respective domains, together in a way that builds on the strengths of each technique and to showcase the viability of the combined SIAR+MPC method in steering unknown game dynamics. We thus believe that this is a novel contribution that could be used by others to either steer game dynamics or to gain insight into more basic steering heuristics/methods as you have mentioned. However, we accept that our current manuscript may not provide these insights.
> > >
> > > 2. To clarify, our method is split into two phases. In the system identification phase, we learn how the system’s dynamics depends on both the state (i.e., the players’ strategies) as well as the control input. Then, during the steering/control phase, we use MPC to continually observe the system’s state and optimize for the optimal control input based on a prediction of the trajectory that comes from the model learnt in the 1st phase. Thus, we do not use SIAR to optimize the control input going forward: there is no “final optimal control sequence”, but rather one that is continually being optimized by MPC during the steering/control phase.
> > >
> > > We are not opposed to other methods for sampling the control input during the system identification phase or for determining a good control sequence during the steering/control phase, but do not at the moment see an easy way to implement the online learning methods you have suggested as which control inputs are good to implement are state dependent (they depend on the players’ current strategies). The control space at one point in the state space does not map readily (in terms of how it affects the dynamics) to the control space at another point in state space, and thus we use MPC in the control phase to continually observe the system and optimize for the control input.

---

### Official Review · Reviewer_VWa9 · 2024-11-04

**Soundness:** 3
**Presentation:** 3
**Contribution:** 2
**Rating:** 6
**Confidence:** 3

**Summary:**

The submission investigates the problem of steering game unknown game dynamics. The submission's approach involves first identifying these dynamics by extending SIAR to control settings. Then, it uses MPC to adjust steer these dynamics. The submission gives examples of its approach for stag hunt, matching pennies and epsilon-rock-paper-scissors.

**Strengths:**

The problem of identifying and steering game dynamics is both difficult and seems to be understudied, though I am not a domain expert. The approach described in the submission seems sensible and technically interesting.

---

Overall, I am not sure how useful the submission is, but I found its object of study and ideas interesting. I consider the latter to be enough to merit acceptance. On the other hand, I am not an expert in this domain, so both my perception of the submission's strengths and weaknesses should be treated with a non-zero amount of skepticism. I left my confidence low so as to leave room for reviewer's who may feel more confident in their expertise on the subject matter.

**Weaknesses:**

1. The submission doesn't do a very good job communicating how the reader ought to be interpreting these incentives.
2. The submissions touts the "diverse range of games" on which it performs experiments. In fact, it performs experiments on 2 2x2 matrix games and 1 3x3 matrix game. I would hesitate to call this diverse.
3. The submission notes the "larger dimensionality" of rock-paper-scissors. This strikes me as somewhat concerning. If the dimensionality of rock-paper-scissors is already noteworthy in the context of the method, is there much hope of applying it to more interesting settings?
4. I think the first paragraph of section 5.2 could be clearer. In the first part of the paragraph, the submission explains that replicator dynamics and learning algorithms with non-vanishing regret possess undesirable behavior. Thereafter, it states "In that regard, ... we demonstrate the performance SIAR-MPC in steering [learning dynamics of non-vanishing regret]." If I am reading between the lines correctly, the submission is meaning to communicate something positive---that it successfully steers learning dynamics with undesirable properties. But the writing doesn't effectively get that point across, in part because there is no previously mentioned "regard" that makes the sentence read correctly.

**Questions:**

How should the reader be interpreting these incentives in the context of the games studied?

How scalable are these approaches to larger settings? Are there fundamental barriers to scaling here or is there hope of overcoming dimensionality-related limitations?

---

> ### Author Response · Authors · 2024-11-23
>
> We are grateful to the reviewer for their insightful feedback. We address the reviewer's concerns below.
>
> **1. The submission doesn't do a very good job communicating how the reader ought to be interpreting these incentives.**
>
> In our framework, incentives are conceptualized as control inputs designed by MPC that affect the payoff matrix. As the payoff matrix, by definition, encodes all the information about the player's incentives, altering the payoffs is the natural method for influencing agent behavior. These incentives can be thought of, for example, as monetary rewards given that certain outcomes occur.
>
> Which payoffs can be affected is a setting-specific question and we experimented this idea in our RPS game where we only allow the controller to add incentives to certain outcomes.
>
> We will make the necessary modifications to ensure these clarifications are appropriately reflected in the relevant sections of the paper.
>
> **2. The submissions touts the "diverse range of games" on which it performs experiments. In fact, it performs experiments on 2 2x2 matrix games and 1 3x3 matrix game. I would hesitate to call this diverse.**
>
> Our intention was to showcase that our method is capable of handling a variety of game types, each with distinct challenges and undesired behaviors. For example, in zero-sum games like Matching Pennies, our method successfully addresses and manages cyclic behaviors, while in Rock-Paper-Scissors, it handles chaotic dynamics effectively. In coordination games, such as Stag Hunt, our approach overcomes the issue of converging to a suboptimal equilibrium, successfully steering the system toward a socially optimal outcome.
>
> Moreover, we conducted additional simulations to demonstrate the performance of the proposed SIAR-MPC method under varying payoff matrices. Specifically, for the Stag Hunt Game, we generated 50 random payoff matrices while ensuring that the game maintained its cooperative nature with a socially optimal equilibrium. We evaluated the performance of SIAR-MPC across 50 systems and compared it with other methods based on our performance metrics, as detailed in Appendix A of the manuscript.
>
> |  Method  |       MSE (Ref.) x₁,₁       |       MSE (Ref.) x₂,₁       |     SS Error x₁,₁      |     SS Error x₂,₁      |       Cost       |
> |:--------:|:---------------------------:|:---------------------------:|:---------------------:|:---------------------:|:----------------:|
> |  SIARc   |      2.01 × 10⁻¹           |      1.80 × 10⁻¹           |      1.80 × 10⁻¹      |      1.80 × 10⁻¹      |    3.71 × 10²    |
> |   PINN   |      5.78 × 10⁻¹           |      5.47 × 10⁻¹           |      5.80 × 10⁻¹      |      5.80 × 10⁻¹      |    1.08 × 10³    |
> |  SINDYc  |      5.38 × 10⁻¹           |      4.87 × 10⁻¹           |      5.52 × 10⁻¹      |      5.36 × 10⁻¹      |    1.70 × 10⁷    |
>
> The results given in the above table clearly demonstrates that SIAR-MPC consistently achieves lower error values and control cost compared to the other methods.
>
> **3. The submission notes the "larger dimensionality" of rock-paper-scissors. This strikes me as somewhat concerning. If the dimensionality of rock-paper-scissors is already noteworthy in the context of the method, is there much hope of applying it to more interesting settings?**
>
> We agree that rock-paper-scissors should not be called a game with "larger dimensionality", and does not showcase our method's scalability. The point of restricting the controllers to be nonzero only on a subset of the outcomes was to demonstrate steerability even with incomplete controls.
>
> **4. I think the first paragraph of section 5.2 could be clearer. In the first part of the paragraph, the submission explains that replicator dynamics and learning algorithms with non-vanishing regret possess undesirable behavior. Thereafter, it states "In that regard, ... we demonstrate the performance SIAR-MPC in steering [learning dynamics of non-vanishing regret]." If I am reading between the lines correctly, the submission is meaning to communicate something positive---that it successfully steers learning dynamics with undesirable properties. But the writing doesn't effectively get that point across, in part because there is no previously mentioned "regard" that makes the sentence read correctly.**
>
> You are reading our intended meaning correctly, and we will make the paragraph clearer in the camera-ready version of the manuscript.
>
> **5. How should the reader be interpreting these incentives in the context of the games studied?**
>
> The incentives in our work should be interpreted as mechanisms like monetary rewards or penalties that are assigned to players with the purpose of influencing their behavior within the game. For example, incentives might take the form of bonuses to encourage cooperative behavior or fines to discourage actions that result in socially suboptimal outcomes. Please see the answer to the Comment 1 above as well.

---

> > ### Author Response · Authors · 2024-11-23
> >
> > **6. How scalable are these approaches to larger settings? Are there fundamental barriers to scaling here or is there hope of overcoming dimensionality-related limitations?**
> >
> > The scalability of  sum of squares techniques is an active area of research, with at least three promising directions. These include leveraging the symmetry or sparsity of the problem, using "easier" cones such as DSOS and SDSOS cones, and use of more scalable alternatives to the interior point methods, such as ADMM. The reviewer can refer to [R2] and the references cited therein for further details.
> >
> > [R2] Anirudha Majumdar, Georgina Hall, and Amir Ali Ahmadi, "Recent scalability improvements for semidefinite programming with applications in machine learning, control, and robotics," Annual Review of Control, Robotics, and Autonomous Systems, vol.3.1, pp. 331-360, 2020.

---

> > > ### Comment · Reviewer_VWa9 · 2024-11-27
> > > **Response**
> > >
> > > Thanks to the authors for their response. My concerns regarding the diversity and dimensionality of games remain, so I will keep my score of marginal accept.

---

### Official Review · Reviewer_BDhm · 2024-11-08

**Soundness:** 3
**Presentation:** 3
**Contribution:** 2
**Rating:** 6
**Confidence:** 2

**Summary:**

This paper propose Side Information Assisted Regression with Model Predictive Control (SIAR-MPC), a framework to learn the dynamics of game and steer game dynamics towards desirable outcomes when data is scarce. This framework has two components, which includes system identification part and MPC part. In system identification step, the algorithm approximated the controlled dynamics using only a limited number of samples. Second, in MPC step, based on the learned dynamics, MPC is applied to steer the system towards a desirable outcome. This framework is evaluated in data-scarce settings and show this framework have superior performance compared to other baselines.

**Strengths:**

- Most of this paper is well-structured and well-written.

**Weaknesses:**

- The font in the plots could be larger, it is relatively hard to read. The introduction of RFI could be more detailed in section 4.1.
- The effectiveness of the algorithm in data-scarce setting could be emphasized more in the experiments, it is interesting to see how the performance is affected when the avalibility of the data varies.

**Questions:**

- what if there is error in the dynamics modelling step, what will happen in the MPC phase? Can MPC accomodate the error?

---

> ### Author Response · Authors · 2024-11-23
>
> We thank the reviewer for their feedback. Please see our responses below.
>
>
> **1. The font in the plots could be larger, it is relatively hard to read.**
>
> We will enlarge the font in the revised manuscript.
>
> **2. The introduction of RFI could be more detailed in section 4.1.**
>
>  We will provide more details for the RFI in the camera-ready version of the manuscript.
>
> **3. The effectiveness of the algorithm in data-scarce setting could be emphasized more in the experiments, it is interesting to see how the performance is affected when the availability of the data varies.**
>
> It is true that this is an interesting question. In our work, we demonstrate that our method is viable for learning and controlling the system in a data-scarce setting even when other methods fail. Exploring results under varying data conditions could be an interesting future work.
>
> **4. What if there is error in the dynamics modelling step, what will happen in the MPC phase? Can MPC accommodate the error?**
>
> Even though MPC is based on a model of the dynamics that we learn during the system-identification step, it does not allow error in the model to propagate unchecked over time as it continually takes in state measurements at each timestep and optimizes for the next few control steps over a relatively small (moving) prediction horizon.
>
>  As the predicted trajectory within this small prediction horizon and the optimal control inputs computed by MPC depend on the model, how close the learned model is to the true dynamics does affect the MPC phase, but in a bounded way.

---

> > ### Comment · Reviewer_BDhm · 2024-11-26
> >
> > Thank you for the response and I will remain my score as it is.

---

### Official Review · Reviewer_mmtn · 2024-11-09

**Soundness:** 2
**Presentation:** 2
**Contribution:** 2
**Rating:** 3
**Confidence:** 2

**Summary:**

This work studies how to steer game dynamics towards desirable outcomes. To do so, the authors introduce a framework that combines side information assisted regression and model predictive control. The framework first tries to perform a system identification step to approximate the control dynamics and subsequently utilizes MPC to steer the system. The authors also give several empirical studies on games, demonstrating the effectiveness of the proposed framework.

**Strengths:**

The problem is well motivated and the paper is overall easy to follow.

**Weaknesses:**

The originality and significance of the contributions seem limited. The framework primarily extends existing techniques, coupled with well-studied MPC approaches. It is not clear how the contributions could be translated into broader insights for the community. Also, is it possible to provide some theoretical justification?

**Questions:**

See above

---

> ### Author Response · Authors · 2024-11-23
>
> We thank the reviewer for their feedback.
>
> While we acknowledge that this work builds on prior research that focuses on system identification for game dynamics, our contributions go beyond this foundation. Previous work primarily focuses on identifying system dynamics without addressing the unique challenges of game dynamics, which often hinder desirable outcomes (as defined in our paper). Our work tackles a fundamentally different problem: steering such dynamics toward desirable outcomes, particularly in data-scarce settings where the underlying dynamics are unknown. While the control method we use is well-established, our novelty lies in its adaptation to incentive-driven game dynamics, offering a fresh perspective and actionable insights for such scenarios.
>
> For the question regarding the theoretical justification:
>
> The SIAR solution is a regressor, and therefore its quality depends on the quality and number of samples. The benefit of SIAR is that it incorporates side-information, and thus reducing the size of the problem's feasible set. To the best of our knowledge, there are no theoretical results that quantify the improvements such a reduction provides to the regressor. A relevant question is whether, a polynomial regressor, as returned by SIAR, can be inherently "bad". In that regard, the results of [R1] can readily be extended in the presence of control, and therefore the existence of a "good" polynomial regressor (a feasible solution) that satisfies the given side-information constraints with SOS certificates is guaranteed to exist for some level of the SOS hierarchy. We will add the proof in the appendix of the revised manuscript.
>
> [R1]  Sakos, Iosif, Antonios Varvitsiotis, and Georgios Piliouras. "Data-Scarce Identification of Game Dynamics via Sum-of-Squares Optimization" arXiv preprint arXiv:2307.06640 (2023).

---

### Note · Authors · 2024-12-02

**Comment:**

We thank all reviewers for their time and effort in reviewing our paper. We have taken the comments onboard and decided to make some improvements and submit to another venue.

**Withdrawal Confirmation:**

I have read and agree with the venue's withdrawal policy on behalf of myself and my co-authors.